# Real-World Data to Assess the Proportion of Patients Admitted for Febrile Neutropenia That Could Be Considered at Low Risk: The Experience of the Centre Hospitalier Universitaire de Québec

**DOI:** 10.3390/curroncol32030133

**Published:** 2025-02-26

**Authors:** Tommy Jean, Camille Sylvestre, Francis Caron, Dominique Leblanc, Geneviève Soucy, Julie Lemieux

**Affiliations:** CHU de Québec, Faculté de Médecine, Université Laval, Quebec, QC G1J 0J9, Canada; tommy.jean.med@ssss.gouv.qc.ca (T.J.); camille.sylvestre.med2@ssss.gouv.qc.ca (C.S.); francis.caron.med@ssss.gouv.qc.ca (F.C.); dominique.leblanc.med@ssss.gouv.qc.ca (D.L.); genevieve.soucy.med1@ssss.gouv.qc.ca (G.S.)

**Keywords:** febrile neutropenia, outpatient, MASCC score, CISNE score

## Abstract

Febrile neutropenia (FN) is a serious complication of chemotherapy that often leads to hospitalization in cancer patients. It is now well-established that carefully selected patients can be safely treated on an outpatient basis. The objective of this study was to assess the number and proportion of patients hospitalized for FN in a university hospital setting who would have met the low-risk criteria for FN, and whether these patients experienced favorable outcomes during hospitalization. We conducted a retrospective study of all patients admitted for FN at three hospitals in Quebec City between 1 January 2018 and 31 December 2019. Patients with leukemia and those who had undergone stem cell transplants were excluded. A retrospective chart review was performed to establish the Multinational Association for Supportive Care in Cancer (MASCC) score for each patient. Based on predefined criteria, we also determined whether the clinical course was favorable or unfavorable. A total of 177 hospitalizations met our inclusion criteria. We found that 101/177 (57.1%) of the hospitalized patients met the low-risk FN criteria according to their MASCC score. Among these, 74/177 (41.8%) met all the criteria for outpatient treatment. The majority of these patients had a favorable outcome (70/74, 94.6%). In contrast, among patients who did not meet the eligibility criteria for outpatient treatment, 44.7% (46/103) experienced favorable outcomes during their hospitalization. These data highlight the importance of patient selection for outpatient care.

## 1. Introduction

Chemotherapy-induced febrile neutropenia (FN) is a serious and frequent complication in patients with cancer. It is a condition that must be treated promptly, and most patients are hospitalized for treatment with broad-spectrum intravenous antibiotics [1,2]. However, morbidity and mortality differ greatly among subpopulations according to several factors (type of neoplasia, age, comorbidity, type of infection, etc.) [3].

It is now well established that carefully selected patients can be safely treated on an outpatient basis [4], and this appears to be the most cost-effective approach. Indeed, Teuffel’s study demonstrated that home treatment for FN, either with intravenous or oral antibiotics, was less costly and more effective compared to inpatient treatment in a Canadian context [5]. The decision to hospitalize has repercussions both on the patient and on the healthcare system. For the patient, this creates significant stress and puts them at risk of nosocomial infection. For the healthcare system, this consumes valuable resources in terms of personnel and materials [5] and accounts for a significant portion of cancer-related costs [6,7].

In this context, several expert groups have incorporated outpatient treatment for the low-risk FN patient population in their guidelines. In 2018, the American Society of Clinical Oncology (ASCO) and the Infectious Diseases Society of America (IDSA) updated practice guidelines to help clinicians identify these patients [8]. This decision is based, first and foremost, on clinical judgment and must consider the entire clinical situation and social context [9]. Validated clinical scores such as the Multinational Association for Supportive Care in Cancer (MASCC), the Clinical Index of Stable Febrile Neutropenia (CISNE), and the Talcott Rules can aid in this decision [10]. The National Comprehensive Cancer Network (NCCN) guidelines go in the same direction [11].

The current practice in many hospitals is still to admit all patients with FN, regardless of their risk level. In this study, we aim to evaluate the proportion of admitted patients that could have been considered at low risk and determine their clinical outcomes. Our hypothesis is that these patients represent a significant number and that the vast majority experience favorable outcomes.

## 2. Materials and Methods

### 2.1. Study Design

We conducted a retrospective cohort study at the *Centre hospitalier universitaire de Québec* (CHU de Québec), a tertiary care university hospital comprising five sites, three of which are hospitals that hospitalize patients with FN. We identified patients admitted for FN as their primary diagnosis on patient discharge summaries from 1 January 2018 to 31 December 2019.

### 2.2. Patients

We included patients aged 18 or older who were diagnosed with solid or hematological neoplasia (excluding acute leukemia, stem cell transplant, and CAR-T cell therapy) and were admitted for FN. FN was defined according to the criteria outlined by the 2018 ASCO/IDSA guidelines as (1) a fever of ≥38.3 °C on one occasion or ≥38.0 °C on two occasions with a minimum interval of one hour and (2) an ANC of <500 × 10^6^/L [5].

We excluded patients with a history of acute leukemia, hematopoietic stem cell transplantation (autologous and allogeneic), or chimeric antigen receptor T (CAR-T) cell therapy; those admitted for reasons unrelated to their FN; patients directed towards palliative care; individuals admitted electively with an incidental diagnosis of FN; and those who had received antibiotic treatment prior to admission.

### 2.3. Objectives

The primary objective was to determine the number and proportion of patients hospitalized for FN who would have met the low-risk criteria according to their MASCC score.

We had two secondary objectives. The first was to assess the proportion of patients considered low-risk according to their MASCC score who met the extended criteria of ASCO, IDSA, and NCCN for outpatient treatment. The second was to identify if, among these low-risk patients, some had experienced unfavorable outcomes during hospitalization.

The criteria for unfavorable outcomes were primarily established based on the criteria from the FINITE study published in 2015 [12]. The following events are considered unfavorable outcomes: death, admission to the intensive care unit, transition to exclusive palliative care, or the occurrence of a major complication. Major complications include acute organ failure (cardiac, renal, or respiratory), delirium, hypotension, major infection, acute abdomen, or serious complications that, in themselves, constitute a reason for hospitalization (e.g., pulmonary embolism, arrhythmia, disseminated intravascular coagulation, or major bleeding) (see Appendix A).

### 2.4. Data Collection

We obtained approval from the hospital to conduct the retrospective chart review. Then, we asked the Medical Records Department to identify all hospitalizations in 2018 and 2019 with FN as the primary diagnosis on the hospital discharge summary form. We reviewed the electronic health records (EHRs) to complete the data collection. We used the *Registre québécois du cancer* (RQC) to confirm the neoplastic diagnosis and its stage.

Variables extracted from the EHRs included patient demographics; length of hospitalization and hospital site; characteristics of neoplasia, including type, stage, and chemotherapy received; biochemical results at admission (renal and liver function and glycemia); parameters for Charlson Comorbidity Index; use of granulocyte colony-stimulating factor (G-CSF) prior to or during hospitalization; initial antibiotic therapy, time to initiation, changes in treatment, and total duration of antibiotics; and results of investigations to identify an infectious source, including imaging and cultures.

#### 2.4.1. MASCC Scoring System

The MASCC score is based on the following variables: severity of symptoms, presence or absence of hypotension, solid or hematological neoplasia with or without a history of previous fungal infections, presence or absence of obstructive pulmonary disease, presence or absence of dehydration requiring intravenous repletion, inpatient or outpatient status before the episode of FN, and age. A result of ≥21 is considered low-risk, while <21 is considered high-risk (see Appendix B for more details).

We assessed the severity of symptoms based on the initial consultation. The presence or absence of hypotension (e.g., systolic arterial blood pressure of <90) was determined based on vital signs in the emergency room. The need for intravenous repletion was considered present if the patient had received a 1 L bolus or >1 mL/kg/hour of fluids.

#### 2.4.2. CISNE Scoring System

The CISNE score is based on the following variables: Eastern Cooperative Oncology Group (ECOG) performance status, stress-induced hyperglycemia (≥6.7 mmol/L or ≥13.9 mmol/L in the case of diabetes or treatment with corticosteroids), COPD diagnosis on therapy (bronchodilators, oxygen, and/or steroids), history of cardiovascular disease (including chronic heart conditions, but excluding a history of a single uncomplicated episode of atrial fibrillation), mucositis grade according to the National Cancer Institute (NCI) score, and monocyte count. A score of 0 points is considered low risk, 1–2 points as intermediate risk, and ≥3 points as high risk (see Appendix B for more details).

To assess the ECOG performance status, we relied on the most recent consultation in the hematology–oncology clinic. In instances where this information was not accessible, we relied on the emergency room consultation. When the score was not clearly indicated, it was inferred from the clinical description provided during the consultation. The latter was also used to ascertain the mucositis grade.

### 2.5. Eligibility Criteria for Outpatient Treatment According to ASCO, IDSA, and NCCN

Patients deemed high-risk according to at least one of these practice guidelines are not eligible for outpatient treatment for FN. Factors considered to be indicative of a high risk include the following: signs and symptoms consistent with severe sepsis, mucositis grade ≥ 3 according to the Common Terminology Criteria for Adverse Events (CTCAE), MASCC score of <21, CISNE score of ≥3, new pulmonary infiltrate or presence of hypoxemia, documented intravascular catheter infection, underlying obstructive lung disease, use of Alemtuzumab in the last 2 months, progressive tumor, and severe renal or hepatic insufficiency [7,9].

We refer to these criteria as the “extended criteria” required to assess a patient’s eligibility for outpatient treatment (see Appendix C for more details).

## 3. Results

From January 2018 to December 2019, a total of 792 patients were hospitalized with a diagnosis of FN. Among these patients, 615 did not meet the inclusion criteria or had at least one exclusion criterion (Figure 1). Specifically, 20 had no history of neoplasia; 489 did not present with FN upon admission; 500 patients had a history of acute leukemia or hematopoietic stem cell transplantation; 66 patients were not primarily admitted for FN; 6 had received antibiotic treatment; and 4 were directly admitted to palliative care.

### 3.1. Patients Characteristics

Of the 177 eligible patients, the mean age was 61.5 years, and more than half (99/177, 55.9%) were women (Table 1). The three most common neoplasia were lymphoma (46/177, 26.0%), breast cancer (35/177, 19.8%), and gastrointestinal cancer (26/177, 14.7%). Regarding solid tumors, the majority were at a localized stage (76/117, 65.0%).

The average Charlson comorbidity index score was 5.8 points. The most frequent comorbidities were uncomplicated diabetes (38/177, 21.5%), complicated diabetes (7/177, 4.0%), a history of myocardial infarction (16/177, 9.0%), chronic obstructive pulmonary disease (12/177, 6.8%), peripheral vascular disease (10/177, 5.6%), stroke or TIA (12/177, 6.8%), moderate to severe renal failure (10/177, 5.6%), and mild liver disease (9/177, 5.1%).

### 3.2. Treatment

The majority of patients were initially treated with meropenem (162/177, 91.5%) (Table 2). The second most commonly used antibiotic was piperacillin–tazobactam (8/177, 4.5%). For 116 patients, the average time between the first medical contact at triage and the first dose of antibiotics was 3.4 h. The total duration of antibiotic therapy averaged 6.9 days, and less than half of the patients had a continuation of antibiotic therapy at home (74/177, 41.8%). G-CSF was administered to the majority of patients during their hospital stay (141/177, 79.7%). The average length of hospitalization was 5.9 days.

### 3.3. Microbiologic Characteristics

The infectious focus was identified in almost half of the patients (83/177, 46.9%). The most common types of infection were pneumonia (28/177, 15.8%), upper respiratory tract infection/sinusitis/tonsillitis (23/177, 13.0%), and catheter-related sepsis (12/177, 6,8%). Blood cultures were positive in a minority of patients (12/177, 6.8%), with Staphylococcus aureus being the most common pathogen (3/177, 1.7%). Staphylococcus aureus was also the most frequently found pathogen across all infectious sites (7/177, 4.0%). For further details, please refer to Appendix D.

### 3.4. Clinical Scores

According to the MASCC score, the majority of patients had a low-risk score (101/177, 57.1%) (Appendix B). Regarding the CISNE score, most patients fell into the intermediate-risk category (101/177, 57.1%) and high-risk category (42/177, 23.7%). Among those initially identified as low-risk according to the MASCC score, 73.3% (74/101) met the criteria for outpatient treatment when the extended criteria were applied, and the majority of these patients had favorable outcomes (70/74, 94.6%) (Table 3 and Figure 2). Of the four patients who experienced unfavorable outcomes, two had bacteremia, one had acute renal failure, and one had delirium. None of them died or were transferred to the intensive care unit. In contrast, among patients who did not meet the eligibility criteria for outpatient treatment, 44.7% (46/103) experienced favorable outcomes during their hospitalization. For further details, please refer to Appendix B.

## 4. Discussion

Identifying FN patients who may be candidates for outpatient treatment remains a significant clinical challenge, especially in emergency room settings. Therefore, clinical scoring systems have been developed to assist clinicians in making decisions.

The MASCC score is a well-known and validated tool for identifying low-risk patients. It was first published in 2000 [13]. Since then, it has been used in several international studies [14]. The score has been validated in heterogeneous populations and applies to patients with both solid and hematological neoplasia [15]. However, the MASCC score is not without its limitations, and up to 11% of low-risk patients have experienced serious complications [2,16]. Subsequently, other risk stratification scores, such as CISNE, have been developed and validated [17]. The CISNE score has only been validated in a population of patients with solid neoplasia who had received mild to moderate chemotherapy treatments and were clinically stable [12]. In these patients, the CISNE score appeared to be superior in assessing overall risk [8].

Although outpatient management for selected patients with FN has been well-established in the guidelines for several years, this practice remains limited in many centers to this day [18]. Clinicians are often concerned about compromising patient safety, primarily due to the challenges associated with risk assessment in a heterogenous population. The incorporation of the extended criteria from ASCO, IDSA, and NCCN allows for the identification of specific conditions that might be overlooked by these scoring systems. In fact, when we combine the MASCC score with the extended criteria, our study shows that they serve as reliable predictors of a favorable clinical outcome, as defined by our pre-established criteria. As previously mentioned, out of the 74 identified patients, only 4 experienced an unfavorable outcome. The proportion of patients potentially treatable as outpatients is, therefore, significant, amounting to 42% of the patients included in our study (74/177) over 2 years. However, if we had included all hospitalizations for FN, the proportion of patients treatable as outpatients would have been considerably lower, at 74/792 (9.3%) over 2 years. Nonetheless, we still believe that 74 hospitalizations over 2 years represents a substantial number of patients for whom hospitalization could have been avoided.

Given the study results, we believe it is relevant to implement outpatient follow-up for low-risk patients with FN, as suggested in the guidelines [8]. A standardized form is being developed to assist emergency room physicians and hematology–oncologists in the initial assessment of patients with FN. This form includes the use of the MASCC score, as well as the extended criteria mentioned in our study that contraindicate outpatient treatment. It is also important to note that, to be eligible for outpatient treatment, a psychosocial assessment must be conducted by the physician. Specifically, the patient must consent to this approach, be compliant with follow-up, have access to a telephone, be able to travel to the hospital in case of deterioration, and be accompanied by a caregiver [8].

To ensure a favorable outcome, close monitoring should be available through regular telephone follow-ups and clinic visits until the episode resolves. The follow-up of neutrophil counts until myeloid recovery will also be included. Patients will be advised to return for follow-up if their fever persists for more than 48 h after starting antibiotics or if there is a recurrence of fever after a period of defervescence. They will be admitted to the hospital if necessary.

Our study has several strengths. Firstly, it addresses a straightforward clinical question with tangible implications for both patients and hospital resource management. The ability to prevent unnecessary hospitalizations provides an advantage for both patients and healthcare facilities. Additionally, our study is grounded in assessment criteria that have been acknowledged and validated in studies and practice guidelines for several years. Lastly, we conducted a thorough review of a substantial number of patient records, which adds value to the study.

Our study also has its limitations. Given its retrospective nature, some score criteria were derived from medical records. For instance, the ECOG performance status in the CISNE score was determined based on clinical impressions derived from reading medical records, allowing room for subjective interpretation. This also applies to calculating the MASCC score, which incorporates the patient’s symptoms at their initial presentation. As the chart review was not blinded, it is possible that our judgment may have been influenced by our knowledge of the patient’s course. For example, when hospitalization is brief and uncomplicated, it could influence assigning a more favorable score to subjective aspects, and conversely, the opposite could be true. This introduces a potential risk of misclassification bias. It is possible that some patients would have been classified as low risk during the initial consultation, but due to an unfavorable progression, the subjective components of the initial clinical scores were retrospectively interpreted in a more unfavorable light. Therefore, we cannot rule out with certainty that the actual proportion of unfavorable outcomes may have been higher than what was found in our study. To address the inherent limitations of a retrospective study, prospective studies would be beneficial to explore this issue. The reliability of clinical scores would be improved.

Furthermore, with prospective studies, the impact of outpatient follow-up on patient quality of life could be explored using validated questionnaires. Few studies have focused on quality of life as a primary outcome [18]. Another limitation is the exclusion of a significant portion of patients with FN from our study, specifically those with leukemia or a history of hematopoietic stem cell transplant. However, in our view, this group is at too high of a risk for complications to be considered for outpatient treatment. These patients undergo more aggressive chemotherapy regimens, face a risk of prolonged neutropenia (exceeding 7 days), and are susceptible to fungemia. These patients have a mortality rate of up to 14.3% during hospitalization for FN [1]. We believe they should not be regarded as potential candidates for outpatient treatment. This perspective aligns with the recommendations provided by ASCO, NCCN, and IDSA through their extended criteria.

Finally, we applied the CISNE score to patients with hematological neoplasia (54/177, %), even though the score is not officially validated for this population. However, the objective of our study was not to verify the concordance between the two scores. The goal was to apply them as broadly as possible to ensure the selection of a low-risk population for which we can apply an outpatient treatment trajectory. As a result, some patients considered low-risk according to the MASCC score may not be deemed suitable candidates for outpatient treatment due to their CISNE score.

## 5. Conclusions

Our study demonstrates that a significant number of patients hospitalized for FN are at a low risk for complications. When these patients are carefully identified according to strict criteria, they generally have favorable outcomes. Outpatient antibiotic treatment within a formalized trajectory could be considered for these patients. Therefore, there is clinical relevance in developing an outpatient monitoring system for these patients in the future.

## Figures and Tables

**Figure 1 curroncol-32-00133-f001:**
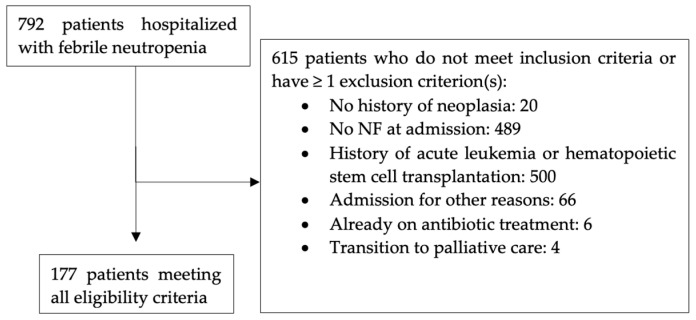
Patient flow chart.

**Figure 2 curroncol-32-00133-f002:**
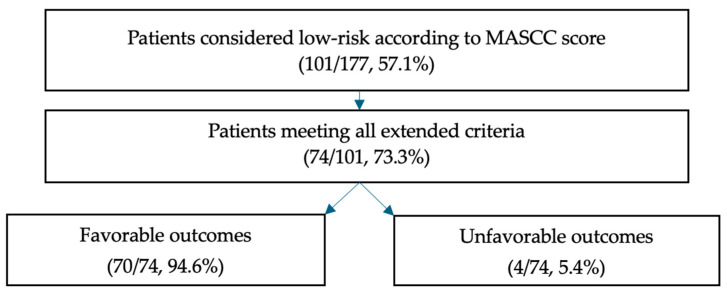
Evolution flow chart.

**Table 1 curroncol-32-00133-t001:** Patient characteristics.

Parameter	Units	Mean (Range)	#/177 (%)
**Age**	Years	61.5 (19–87)	
**Sex**			
Male			78 (44.1)
Female			99 (55.9)
**Neoplasia**			
Lymphoma			46 (26.0)
Breast			35 (19.8)
Gastrointestinal			26 (14.7)
Lung			16 (9.0)
Urinary tract			13 (7.3)
Gynecological			12 (6.8)
Sarcoma			12 (6.8)
MDS and MPS ^1,2^			8 (4.5)
Neuroendocrine			5 (2.8)
Head and neck			2 (1.1)
Others ^3^			2 (1.1)
**Charlson score**	Points	5.8 (2–15)	
Age	<50 years (0)		32 (18.1)
	50–59 years (+1)		30 (16.9)
	60–69 years (+2)		47 (26.6)
	70–79 years (+3)		59 (33.3)
	80 years (+4)		9 (5.1)
Myocardial infarction	Positive (+1)		16 (9.0)
Congestive heart failure	Positive (+1)		4 (2.3)
Peripheral vascular disease	Positive (+1)		10 (5.6)
Stroke or TIA ^4^	Positive (+1)		12 (6.8)
Major neurocognitive disorder	Positive (+1)		0 (0.0)
CPOD ^5^	Positive (+1)		12 (6.8)
Collagenosis	Positive (+1)		2 (1.1)
Peptic ulcer	Positive (+1)		2 (1.1)
Hepatic disease	Mild (+1)		9 (5.1)
	Moderate to severe (+3)		0 (0.0)
Diabetes	Uncomplicated (+1)		38 (21.5)
	With organ involvement (+2)		7 (4.0)
Hemiplegia	Positive (+2)		0 (0.0)
Kidney failure	Moderate to severe (+2)		10 (5.6)
Solid neoplasia	Localized (+2)		76 (43.0)
	Metastatic (+6)		47 (26.6)
Leukemia	Positive (+2)		0 (0.0)
Lymphoma	Positive (+2)		46 (26.0)
AIDS ^6^	Positive (+6)		0 (0.0)

^1^ MDS (Myelodysplastic Syndromes). ^2^ MPS (Myeloproliferative syndromes). ^3^ Carcinoma of unknown origin and mixed carcinoma. ^4^ Transient ischemic attack. ^5^ Chronic pulmonary obstructive disorder. ^6^ Acquired immunodeficiency syndrome.

**Table 2 curroncol-32-00133-t002:** Summary of treatment.

Parameter	Units	Mean (Range)	#/177 (%)
**Initial antibiotic**			
Meropenem			162 (91.5)
Piperacillin–tazobactam			8 (4.5)
Ceftriaxone			3 (1.7)
Levofloxacin			2 (1.1)
Ceftazidime			1 (0.6)
Vancomycin			1 (0.6)
**Time between first medical contact and first antibiotic dose**	Hour(s)	3.4 (0.25–18.5) *	
**Total duration of antibiotic therapy**	Day(s)	6.9 (1–42) *	
**Oral relay at discharge**	Yes		74 (41.8)
**Time between first antibiotic dose and last temperature spike of ≥38.3 °C**	Hour(s)	17.8 (0–267) *	
**Use of G-CSF ^1^ before hospitalization**	Yes		60 (33.9)
**Use of G-CSF during hospitalization**	Yes		141 (79.7)

* The information was available for 116, 173, and 118 respectively. ^1^ Granulocyte colony-stimulating factor.

**Table 3 curroncol-32-00133-t003:** Unfavorable evolution criteria.

Parameter	Units	Mean (Range)	#/177 (%)
**Death**			5 (2.8)
**Admission to the intensive care unit**			13 (7.3)
Length of stay in ICU	Day(s)	6.8 (2–23)	
**Referral to palliative care**			6 (3.4)
**Acute organ failure**	Yes		24 (13.6)
Renal			7 (4.0)
Cardiac			10 (5.6)
Pulmonary			12 (6.8)
**Delirium**	Yes		10 (5.6)
**Hypotension**	Yes		18 (10.2)
**Major infection**	Yes		41 (23.2)
**Serious complication**	Yes		3 (1.7)
Pulmonary embolism			1 (0.6)
Arrythmia			1 (0.6)
Disseminated intravascular coagulation			2 (1.1)
Major bleeding			0 (0.0)
**Acute abdomen**	Yes		0 (0.0)
**1 criterion**			61 (34.5)
**0 criteria**			116 (65.5)

## Data Availability

We do not have any additional data.

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
