# Peer review of "Real-World Data to Assess the Proportion of Patients Admitted for Febrile Neutropenia That Could Be Considered at Low Risk: The Experience of the Centre Hospitalier Universitaire de Québec"

_curroncol, 2025, doi:10.3390/curroncol32030133_

Round 1
Reviewer 1 Report
Comments and Suggestions for Authors
The aim of this retrospective single-centre (tertiary care university hospital) study is to asses the proportion of cancer patients with febrile neutropenia that could be considered at low risk, and thus could be treated on the outpatient basis.
The most serious flaw of the study is the unreliable recruiment of the data. As study has been submitted now, and corresponds to the period 2018-2019, it is not reliable to base the research on informations/data collected on, as the authors stated several times, "on the severity of symptoms based on initial consultation", "on the clinical impression derived from reading the medical records", "on the most recent consultation in hematology-oncology clinic", or "on the emergency room consultation".
The main criteria for risk assesment - severity and duration of neutropenia - is missing; besides, monocyte count and inflammatory markers are very helpful. Thus, the variables need to be completed, and chemotherapy regimen added.
The other limitation is the exclusion of acute leukemia patients; in this case the calculated proportion of cancer patients with febrile neutropenia is not adequate. Besides, in 2.2. inclusion of patient "with solid tumors and hematologic neoplasia" is incorrect, as leukemias are hematologic malignancies.
Among other score, the authors used CISNE score, which has not been validated out of solid tumors, and in this study patients with lymphoma are also included.
Lymphoma is a broad category; non-Hodgkin lymphoma (NHL) and Hodgkin lymphoma are diverse categoris, and should be separated. Some NHLs are treated similar to acute leukemias.
When listing acute complications, terms "major infection" and "serious complications" should be clarified.
The term "progressive tumor in the patient" (line 132) should be corrected as tumor cannot be out of patient; pneumonia is not "the syte of infection" but the type (line 174), one abbreviated "febrile neutropenia" is FN (eg, line 195), "external" antibiotic treatment (line 251) substituted with the outpatient setting.
Figure 1 and Table 2 should be deleted; the description in the text is sufficient.
Comments on the Quality of English LanguageRequires minor editing.
Author Response
Thank you very much for taking the time to review this manuscript. Please find the detailed responses below and the corresponding revisions/corrections highlighted/in track changes in the re-submitted files.
Comments 1: The aim of this retrospective single-centre (tertiary care university hospital) study is to asses the proportion of cancer patients with febrile neutropenia that could be considered at low risk, and thus could be treated on the outpatient basis.
Response 1: We thank you for your insightful comments.
Comments 2: The most serious flaw of the study is the unreliable recruiment of the data. As study has been submitted now, and corresponds to the period 2018-2019, it is not reliable to base the research on informations/data collected on, as the authors stated several times, "on the severity of symptoms based on initial consultation", "on the clinical impression derived from reading the medical records", "on the most recent consultation in hematology-oncology clinic", or "on the emergency room consultation".
Response 2: We are aware that the retrospective nature of the study introduces certain limitations that are inherent to the methodology. Some components of clinical scores, such as MASCC and CISNE, are more subjective. Since these scores were not calculated in most cases at the time of the initial consultation, some of their components were inferred from medical notes. For example, the "Burden of illness" component of the MASCC score and the ECOG score in the CISNE were determined based on the medical notes. For example, a patient that came only for fever with no other symptoms was classified in the MASCC score, in the “burden of symptoms” as “no or mild symptoms” and given 5 points. We have highlighted this limitation in the discussion section. We acknowledge that this introduces a possible bias, which has been addressed in the revised version of our manuscript.
Comments 3: The main criteria for risk assesment - severity and duration of neutropenia - is missing; besides, monocyte count and inflammatory markers are very helpful. Thus, the variables need to be completed, and chemotherapy regimen added.
Response 3: Regarding the duration of neutropenia, we excluded patients who were at high risk to develop severe prolonged neutropenia, such as those with acute leukemia, those who had undergone a stem cell transplant, or those who received CAR-T cell therapy.
As for monocytes, we calculated the CISNE score, which includes the value of monocytes. For the sake of conciseness, we chose not to present this data individually but they were used in the score.
Indeed, it would have been interesting to assess inflammatory parameters. However, this information was not collected, as it is not part of the clinical scores typically used to evaluate the risk of febrile neutropenia. Moreover, the IDSA guidelines specify that there is currently insufficient data to recommend the routine use of C-reactive protein or procalcitonin. Since these elements are not part of the recommended initial evaluation, we did not include them in our data collection.
Concerning chemotherapy regimens, the data were collected, but again, we decided not to present them to maintain brevity. If the editors consider it relevant to include this information, we would be pleased to provide it in an appendix.
Comments 4: The other limitation is the exclusion of acute leukemia patients; in this case the calculated proportion of cancer patients with febrile neutropenia is not adequate. Besides, in 2.2. inclusion of patient "with solid tumors and hematologic neoplasia" is incorrect, as leukemias are hematologic malignancies.
Response 4: As for the exclusion of patients with leukemia, we made this decision as we believe these patients are at too high a risk of complications to consider outpatient treatment. The same applies to patients who have undergone stem cell transplantation or CAR-T therapy. We have noted in the discussion that these represent a significant proportion of patients admitted for febrile neutropenia. However, they were not part of the low-risk population we intended to study.
Following your insightful comment, we have revised the manuscript and clarified in section 2.2 “with solid or hematological neoplasia (excluding acute leukemia, stem cell transplant, CAR-T cell therapy)”
Comments 5: Among other score, the authors used CISNE score, which has not been validated out of solid tumors, and in this study patients with lymphoma are also included.
Response 5: Regarding the use of the CISNE score for hematologic malignancies, we mention in the discussion that the score is not validated in this context. However, the objective of our study was not to verify the concordance between the two scores. The goal was to apply them as broadly as possible to ensure the selection of a low-risk population. In the revised manuscript, we clarified that this point.
Comments 6: Comments Lymphoma is a broad category; non-Hodgkin lymphoma (NHL) and Hodgkin lymphoma are diverse categoris, and should be separated. Some NHLs are treated similar to acute leukemias.
Response 6: For the lymphoma categories, it is indeed true that this is a diverse category. However, we excluded patients who had undergone stem cell transplantation, CAR-T therapy, or prior treatment with alemtuzumab.
Comments 7: When listing acute complications, terms "major infection" and "serious complications" should be clarified.
Response 7: This is also an excellent comment. In the revised version of the manuscript, we have included Appendix A, which provides a detailed list of all major complications. These criteria are based on the FINITE study, which validated the CISNE score.
Comments 8: The term "progressive tumor in the patient" (line 132) should be corrected as tumor cannot be out of patient; pneumonia is not "the syte of infection" but the type (line 174), one abbreviated "febrile neutropenia" is FN (eg, line 195), "external" antibiotic treatment (line 251) substituted with the outpatient setting.
Response 8: We appreciate your grammatical suggestions. As non-native English speakers, we strive to write to the best of our ability. We have revised the text of our manuscript following your suggestions.
Comments 9: Figure 1 and Table 2 should be deleted; the description in the text is sufficient.
Response 9: We can move Figure 1 and Table 2 to the appendix without any issue.

Reviewer 2 Report
Comments and Suggestions for Authors
The article “Real-world data to assess the proportion of patients admitted for febrile neutropenia that could be considered at low risk: the experience of the Centre Hospitalier Universitaire de Québec” is interesting, I have following comments/suggestions,
1. Abstract: please define MASCC.
2. It is not clear why the authors have reported a six-year-old data.
3. The introduction is well written, and the aims are clear.
4. The methods are adequately described but the authors are advised to define the used abbreviations.
5. The results are interesting and novel.
Author Response
Thank you very much for taking the time to review this manuscript. Please find the detailed responses below and the corresponding revisions/corrections highlighted/in track changes in the re-submitted files.
Comments 1: The article “Real-world data to assess the proportion of patients admitted for febrile neutropenia that could be considered at low risk: the experience of the Centre Hospitalier Universitaire de Québec” is interesting, I have following comments/suggestions,
1. Abstract: please define MASCC.
Response 1: We thank the reviewer. We have addressed the missing abbreviations, including MASCC score in the abstract.
Comments 2: It is not clear why the authors have reported a six-year-old data.
Response 2: Regarding the timeline, we are aware that the data were collected a few years ago. The project initially began as a resident’s initiative. However, we believe the results remain relevant and applicable to current clinical practice. This study served as a foundation for initiating the development of a program to monitor patients with febrile neutropenia in an outpatient setting at the CHU de Québec.
Comments 3: The introduction is well written, and the aims are clear.
Response 3: We thank the reviewer.
Comments 4: The methods are adequately described but the authors are advised to define the used abbreviations.
Response 4: We thank the reviewer. We revised the manuscript to ensure that abbreviations were used correctly.
Comments 5: The results are interesting and novel.
Response 5: We thank the reviewer.
Reviewer 3 Report
Comments and Suggestions for Authors
The article presents a relevant and focused study addressing the clinical challenge of identifying low-risk febrile neutropenia (FN) patients eligible for outpatient treatment. This is a timely topic with significant implications for optimizing patient care and resource utilization. The study’s use of retrospective chart reviews and validated scoring systems, such as MASCC and CISNE, provides a solid methodological foundation. Its clear objectives and significant findings—demonstrating that a notable proportion of patients can safely avoid hospitalization—are both impactful and practical for influencing clinical guidelines. Furthermore, the authors demonstrate transparency by acknowledging the study's limitations, which strengthens its credibility.
Despite its strengths, the article has areas for improvement. While the results are robust, they would benefit from greater contextualization within the broader literature, particularly through comparisons to similar studies regarding outpatient eligibility rates and clinical outcomes. A deeper exploration of cost implications, which would add practical value to the findings, is also recommended.
-Although the authors acknowledge potential biases due to the study's retrospective nature, more detailed discussions on how these biases might have influenced outcomes—and how future studies could mitigate them—would be beneficial.
- Expanding the abstract to include key numerical findings, such as the proportion of low-risk patients and their outcomes, would also improve its informativeness.
-More detailed descriptions of how subjective measures like ECOG performance status were standardized would enhance the clarity of the methodology.
-Visual elements, such as bar graphs or enhanced figures, could further emphasize key trends and improve the article’s readability.
-While the article is well-written, minor grammatical and typographical refinements would enhance its presentation. Ethical considerations, while briefly mentioned, could be elaborated upon to strengthen the study's rigor. Additionally, the discussion section would benefit from clearer recommendations for implementing outpatient care pathways based on the findings. Including suggestions for prospective studies and incorporating patient-reported outcomes would provide a more comprehensive understanding of outpatient care’s quality and safety.
Addressing these suggestions would further enhance its impact, readability.
Comments on the Quality of English Languagehile the article is well-written, minor grammatical and typographical refinements would enhance its presentation.
Author Response
Thank you very much for taking the time to review this manuscript. Please find the detailed responses below and the corresponding revisions/corrections highlighted/in track changes in the re-submitted files.
Comments 1: The article presents a relevant and focused study addressing the clinical challenge of identifying low-risk febrile neutropenia (FN) patients eligible for outpatient treatment. This is a timely topic with significant implications for optimizing patient care and resource utilization. The study’s use of retrospective chart reviews and validated scoring systems, such as MASCC and CISNE, provides a solid methodological foundation. Its clear objectives and significant findings—demonstrating that a notable proportion of patients can safely avoid hospitalization—are both impactful and practical for influencing clinical guidelines. Furthermore, the authors demonstrate transparency by acknowledging the study's limitations, which strengthens its credibility.
Response 1: We thank the reviewer.
Comments 2: Despite its strengths, the article has areas for improvement. While the results are robust, they would benefit from greater contextualization within the broader literature, particularly through comparisons to similar studies regarding outpatient eligibility rates and clinical outcomes. A deeper exploration of cost implications, which would add practical value to the findings, is also recommended.
Response 2: Regarding the contextualization of the study within a broader body of literature, we completely agree with your suggestion. Based on your feedback, we have updated our manuscript. Specifically, we mention a Canadian study that specifically examined the cost-benefit of outpatient management of febrile neutropenia.
Comments 3: Although the authors acknowledge potential biases due to the study's retrospective nature, more detailed discussions on how these biases might have influenced outcomes—and how future studies could mitigate them—would be beneficial.
Response 3: This is also an excellent comment. Following it, we revised our discussion to emphasize the influence of biases introduced by the retrospective nature of the study. We also addressed the potential impacts on our results in the revised version of our manuscript.
Comments 4: Expanding the abstract to include key numerical findings, such as the proportion of low-risk patients and their outcomes, would also improve its informativeness.
Response 4: The abstract has been updated to include all relevant results mentioned in the study.
Comments 5: More detailed descriptions of how subjective measures like ECOG performance status were standardized would enhance the clarity of the methodology.
Response 5: As for the ECOG score, it was determined during the most recent consultation at the hematology-oncology clinic. In cases where this information was not available, we relied on the emergency room consultation. The CISNE score, which includes the ECOG, distinguishes between scores below 2 and above 2. Based on the description provided during the consultation, the score was inferred. For instance, a patient described as fully ambulatory and in good general condition was assigned a score of 0 or 1. Conversely, a bedridden patient was assigned a score of 2 or 3. However, this remains an interpretation based on retrospective notes. We have acknowledged this limitation in the discussion.
Comments 6: Visual elements, such as bar graphs or enhanced figures, could further emphasize key trends and improve the article’s readability.
Response 6: We thank you for your insightful comment. We have added Figure 2 to enhance the reader's understanding.
Comments 7: While the article is well-written, minor grammatical and typographical refinements would enhance its presentation. Ethical considerations, while briefly mentioned, could be elaborated upon to strengthen the study's rigor. Additionally, the discussion section would benefit from clearer recommendations for implementing outpatient care pathways based on the findings. Including suggestions for prospective studies and incorporating patient-reported outcomes would provide a more comprehensive understanding of outpatient care’s quality and safety.
Addressing these suggestions would further enhance its impact, readability.
Response 7: We apologize for the grammatical errors. French is our native language, and we strive to write in English to the best of our ability.
We have adjusted the Ethics section by detailing the measures taken during our study.
In response to your comments, we have revised the discussion to expand on the outpatient follow-up program for patients with febrile neutropenia. Additionally, we have included suggestions for future studies that would be relevant.

Round 2
Reviewer 2 Report
Comments and Suggestions for Authors
The authors have addressed all of my comments/suggestionsin their revised submission.
Reviewer 3 Report
Comments and Suggestions for Authors
The article has been improved as expected.